# The Role of Radiotherapy in Treating Kaposi’s Sarcoma in HIV Infected Patients

**DOI:** 10.3390/cancers14081915

**Published:** 2022-04-10

**Authors:** Laurent Quéro, Romain Palich, Marc-Antoine Valantin

**Affiliations:** 1INSERM U1160, Alloimmunity-Autoimmunity-Transplantation Research Unit, Université Paris Cité, 75006 Paris, France; 2Radiation Oncology Department, DMU ICARE, Saint Louis Hospital, AP-HP, 75010 Paris, France; 3Tropical Medicine and Infectious Disease Department, Pitié-Salpêtrière Hospital, AP-HP, 75013 Paris, France; romain.palich@aphp.fr (R.P.); marc-antoine.valantin@aphp.fr (M.-A.V.)

**Keywords:** HIV, AIDS, Kaposi, radiotherapy

## Abstract

**Simple Summary:**

Radiation therapy is highly effective and well tolerated in the treatment of localized Kaposi’s sarcoma. In this work, we reviewed the literature to evaluate the efficacy and side effects of radiotherapy in Kaposi’s sarcoma before and after the initiation of highly active antiretroviral therapy and described the indications and modalities of radiotherapy treatment.

**Abstract:**

Kaposi’s sarcoma (KS) is a radiosensitive cancer regardless of its form (classical, endemic, AIDS-related, and immunosuppressant therapy-related). Radiotherapy (RT) is an integral part of the therapeutic management of KS. RT may be used as the main treatment, in the case of solitary lesions, or as palliative therapy in the disseminated forms. The dose of RT to be delivered is 20–30 Gy by low-energy photons or by electrons. The complete response rate after RT is high, around 80–90%. This treatment is well tolerated. However, patients should be informed of the possible risk of the development of late skin sequelae and the possibility of recurrence. With the advent of highly active antiretroviral therapy (HAART), the indications for RT treatment in HIV-positive patients have decreased.

## 1. Introduction

Kaposi sarcoma (KS) is an endothelial cells malignancy caused by human herpesvirus 8 (HHV-8), also known as Kaposi’s sarcoma-associated herpesvirus (KSHV). KS can affect the skin, lymph nodes, and/or viscera. The skin lesions may present in different clinical forms such as ulcerations, patches, plaques, or colored nodules (red, bluish, or sometimes brown), single or multiple, and are frequently associated with edema. These lesions can be painful and cause bleeding or infection such as cellulitis. Histologically, KS is characterized by a combination of an inflammatory infiltrate, endothelial cells, and spindle-shaped cells. In HIV-infected patients, KS is classified by the Centers for Disease Control and Prevention (CDC) as an AIDS-defining malignancy. With the development of highly active antiretroviral therapy (HAART), the incidence of KS has greatly decreased in the US, from a peak of about 47 per million in the 1990s to 6 per million currently.

KS’s high radiosensitivity has been known since 1900, with the first cases of radiotherapy’s efficacy in treating KS [1].

RT has thus become an integral part of the therapeutic management of KS. RT may be used as the main treatment, in case of solitary lesions, or as palliative therapy in the disseminated forms. Radiotherapy is effective in treating both skin and mucosal lesions. According to NCCN guideline [2], radiotherapy is indicated in case of symptomatic and/or cosmetically unacceptable cutaneous limited lesion. RT is an effective treatment for relief of pain, stopping the bleeding, and reducing swelling. In case of extended disease, RT allows an improvement of the symptoms, of the aesthetic aspect of the lesions, and of the quality of life while being very well tolerated.

The optimal dose of radiation to deliver is not clearly defined. KS is sensitive to moderate doses of radiation: most studies have used doses between 20 Gy and 30 Gy. Overall tumor response rate reported from previously published studies was from 47% to 99% [3]. It has been reported that all forms of KS (classical or sporadic, endemic (African), epidemic (AIDS-related), and iatrogenic (immunosuppressant therapy-related)) have the same radiosensitivity. In a retrospective study performed by Caccialanza et al., on 711 lesions of classic KS and 771 lesions of human immunodeficiency virus (HIV)-related KS, treated with radiotherapy, a complete response rate of 98.7% and 91.43% was observed, respectively [4].

The therapeutic management of KS is based on an individual approach which takes into account the criteria of extension of the disease, the localized or disseminated character of the lesions, the criteria predicting the evolution of the disease, in particular the immunovirological situation of the patients, and the possible comorbidities.

Different types of local treatments are available, such as radiotherapy, cryotherapy, surgery, cream application (imiquimod, alitretinoin), and intralesional injection of vinblastine. The treatment options include mainly local and systemic treatments such as chemotherapy and immunomodulatory drugs [2]. In a single-center retrospective study of 160 patients followed for endemic and classical KS, Benajiba et al. reported that 14% of patients did not require any treatment while 45% and 41% of them required local or systemic treatments, respectively. Local treatment consisted of surgery (36%), local chemotherapy (30%), radiotherapy (26%), or other (8%). Systemic treatments included low dose interferon (50.0%), chemotherapy (taxanes or anthracyclines-based regimens) (46%), or other therapies (5%) [5].

## 2. Treatment Modalities

For cutaneous or superficial tumors, different irradiation techniques are possible: contact radiotherapy using low-energy X-ray photons for departments equipped with a dedicated machine; for other departments, an electron beam with a bolus will be used. The energy of the radiation will depend on the thickness of the tumor to be treated.

For head and neck or visceral tumors, conformal radiotherapy/IMRT using high energy photons from a linear accelerator will be used.

The radiation field will include not only the visible part of the tumor, but also a possible associated infiltrated component. The peritumoral margin to be added to the macroscopic lesion is about 10 mm. The European Dermatology Forum (EDF), the European Association of Dermato-Oncology (EADO), and the European Organization for Research and Treatment of Cancer (EORTC)’s guidelines recommend delivering doses between 30–36 Gy in 2- or 3-Gy daily fractions using orthovoltage (low energy) X-ray or electrons beam therapy 3. In palliative treatment, a single radiation dose of 8 Gy has been reported to be associated with very good antitumor efficacy [6].

### 2.1. Randomized Trials

Two randomized studies have evaluated different irradiation regimens in HIV-related KS patients. Singh et al. compared a 24 Gy in 12 fractions radiation regimen (standard arm) vs. 20 Gy in 5 fractions (experimental arm) in 60 patients who were treated for a total of 65 sites (35 sites in the standard arm and 30 sites in the experimental arm) [7]. In this study, the two treatment regimens were non-statistically different regarding treatment response, local recurrence-free survival, and toxicity. The objective and complete response rates were 91% vs. 96% and 57% vs. 56%, respectively. The mean time to maximum objective response was 3 months. Caution was advised by the authors when using the shorter regimen of 20 Gy in five fractions for the treatment of very extensive lesions and in those patients with very severe lymphoedema, as four patients receiving this regimen developed ulceration or necrosis.

Stelzer et al. compared three radiotherapy regimens: 40 Gy in 20 fractions (23 lesions), 20 Gy in 10 fractions (24 lesions), and 8 Gy in 1 fraction (24 lesions). Authors reported that complete response was significantly higher (*p* = 0.04) in patients who received 40 Gy (83%) or 20 Gy (79%) than those who received 8 Gy (50%) [8]. Lesion failure was lower (*p* = 0.03) with 40 Gy (52%) than with 20 Gy (67%) or 8 Gy (88%). Median time to failure was 43 weeks with 40 Gy, 26 weeks with 20 Gy, and 13 weeks with 8 Gy (*p* = 0.003). Out of the seven recurrent lesions that were re-irradiated, six were successfully re-irradiated with a complete clinical response.

Acute and late toxicities were limited to grade 1 and consisted of skin erythema, dry desquamation, alopecia, or hyperpigmentation. The incidence of acute toxicity increased with increasing dose (*p* = 10^−8^). Late toxicity was observed only at 40 Gy (*p* = 0.0007). There were no disruptions of radiation treatment course due to toxicity.

### 2.2. Results of Radiotherapy before the HAART Era

De Wit et al. reported a retrospective study of 31 patients with HIV-related-KS who had received 74 radiation treatments by orthovoltage or megavoltage X-ray (Table 1). The objective response rate was 34% with a subjective lesion improvement in 90% of cases after using a single radiation dose of 8 Gy [9].

Between 1983 and 1990, before the era of ART, Wulf et al. reported a very large retrospective monocentric study from Denmark, the efficacy of low dose irradiation in 2305 AIDS-related KS [16]. Seventy-four skin KSs were treated in three sessions of 2 Gy, one session every 2 weeks. The rate of success was 70%. A subsequent cohort of 2066 patients with skin KSs and 165 patients with mucosal KSs received 3 sessions of 4 Gy, one session every 2 weeks, with a very high success rate of 93% and 91%, respectively, and mean time to relapse of 32 weeks (range = 9–76 weeks) and 21 weeks on average (range = 9–48 weeks). Regarding efficacy on skin lesions, the 4 Gy regimen was significantly superior to the 2 Gy regimen as initial treatment (*p* < 0.0001). Recurrences were found in 6 patients after treatment with the 2 Gy regimen. After retreatment with the 4 Gy regimen, no recurrences were observed with a mean follow-up of 159 weeks (range = 23–190 weeks). Four mucosal KSs (out of 15) that had recurred were retreated and observed for an average of 58 weeks (range = 6–98 weeks) without recurrence.

In a French single-center retrospective study of 420 HIV-positive patients with KS treated with radiotherapy, 186 were treated for lesions of the oral cavity, eyelids, or conjunctiva or penis/scrotum [19]. Ophthalmic KS lesions appeared to be more radiosensitive than skin lesions with an objective remission rate of 96% (98/102 patients) after moderate irradiation doses from 10 to 20 Gy. Penile/scrotal lesions also had a good response rate to moderate dose irradiation, with 34 of 49 patients having a complete response (69.4%). Severe, moderate, and mild reactions were observed in 6 (22%), 4 (15%), and 17 (63%), respectively.

From 1986 to 1996, 643 patients presenting with AIDS-related KS were treated with irradiation at Mondor University Hospital in France. Radiotherapy was delivered by electron beam for extended cutaneous disease, megavoltage X-ray for oral tumor, and orthovoltage X-ray for localized lesions. The delivered dose was 20 Gy in 2 weeks (2.5 Gy/fraction, 4 fractions/week) followed by 2 weeks rest and second series of 10 Gy in 1 week. For oral cavity lesions, the delivered dose was 15.2 Gy in 2 weeks (1.9 Gy/fraction, 4 fractions/week), +/− followed by a similar second series of 15.2 Gy after a 3 weeks rest [15]. After treatment, the objective response rate was 92% among the 621 patients who were evaluable.

### 2.3. Alternative Treatment to Radiotherapy

Cryotherapy could represent an alternative to radiotherapy in some indications. Cryotherapy is a palliative treatment option for small (less than 1 cm), superficial, or patchy lesions of KS. Unlike surgery, cryotherapy, in this indication, does not require anesthesia and does not cause scarring. Kutlubay et al. have reported, in a single-center retrospective study, the efficacy of cryotherapy treatment (spray and/or probe) on 135 KS lesions in 30 patients. The mean number of lesions was 3.2 per patient and the mean size was 0.94 cm (range = 0.3–3 cm). A complete response was observed in 19 out of 30 patients (63%). No complications were observed [20].

### 2.4. Plesiobrachytherapy

Plesiobrachytherapy is a brachytherapy technique that consists of applying a gel plate containing catheters through which a radioactive source will travel to the area to be irradiated. 

It could be an interesting technique of irradiation to treat extensive skin lesions on non-planar surfaces such as the legs, the arms, or the fingers [21].

### 2.5. Results of Radiotherapy from the Beginning of the HAART Era

Due to the improved immune function with the use of HAART, the incidence of KS has dramatically decreased in developed countries since 1996–1997 [22]. Despite its lower incidence, KS remains the most frequent tumor in HIV-infected patients worldwide. In more than 50% of cases, the skin lesions are associated with a more or less important visceral involvement, particularly to the oral cavity and the gastrointestinal in 35% and 40% of cases, respectively [23]. In a prospective cohort study from the United Kingdom, Bower et al. reported 5-year overall survival of 95% and progression-free survival of 77% after treatment with HAART alone in 213 HAART-naive patients with T0 KS, and 5-year overall survival was 85% after treatment with HAART and liposomal anthracycline chemotherapy in 140 patients with T1 KS (Table 1). In this study, only 10 patients out of 303 with T0 KS (HAART naive and non-naive) had received radiotherapy in combination with HAART [24].

In a retrospective study of 18 patients with 37 HIV-related KS lesions treated by radiotherapy delivering a dose between 20 Gy and 36 Gy, Donato et al. observed a complete response rate of 84%, a partial response rate of 16%, and no tumor progression [17]. Radiotherapy was well tolerated with only grade 1 skin or mucosal acute toxicity according to the RTOG scale. There was no radiotherapy cessation due to treatment toxicity. Good cosmetic results were reported in 25 lesions (66%) out of 31 with complete response. Effective palliative effect was obtained for all lesions except for two (5%), located in a vertebra and hard palate.

In a retrospective study of 17 patients with 97 KS skin sites treated by 1 × 8 Gy, 5 × 4 Gy or 10 × 3 Gy radiotherapy, Tsao et al. reported the outcome of 3 patients with 8 HIV-related KS lesions. Two patients with 1 KS site experienced a complete tumor response and 1 patient with 6 KS sites had a partial tumor response at each tumor site [18]. The most common side effects were dry desquamation, hyperpigmentation, and lymphedema.

## 3. Radiotherapy Tolerance

Because KS is a relatively radiosensitive disease, radiotherapy is prescribed with a moderate total dose and is thus generally well tolerated, with mainly grade 1–2 toxicities (Table 1). However, given the sensitivity of mucous membranes to irradiation in HIV-positive patients, irradiation of the conjunctiva and the upper aerodigestive tract should be conducted with caution.

## 4. Outcome after Radiotherapy

KS lesions may take up to 4 months to regress clinically, while edema may take up to 6–12 months to regress. In case of recurrence, a complete response after re-irradiation is rarely observed, and a partial response can be observed in half of the cases and a progression in the other half [10]. Radiotherapy is well tolerated. However, patients should be informed of the possible development of late skin sequelae such as telangiectasia, hyperpigmentation, skin atrophy, fibrosis, and/or hair loss. Radiotherapy leads the tumor and/or the skin infiltration to regress; the blue pigmentation of the lesions will turn black or become depigmented. Patients should also be informed of a possible risk of recurrence outside the radiation field, hence the need for the patient to be regularly examined by a dermatologist. Radiotherapy is a loco-regional treatment which is part of a multimodal management strategy of Kaposi’s disease; its only aim is the control of the disease in the irradiated areas.

## 5. Conclusions

In the HAART era, radiation therapy is still an integral part of the treatment strategy for HIV-infected patients with isolated skin/mucosal lesions or symptomatic disseminated disease. Radiation therapy is an effective treatment for HIV-infected patients, with a high complete response rate while being well tolerated.

## Figures and Tables

**Table 1 cancers-14-01915-t001:** Studies of radiotherapy for Kaposi’s Sarcoma in HIV-infected patients.

Authors	Year of Publication	Country	Number of Patients	Number of Treated Lesions	RT Dose	Overall Response Rate	CR Rate	Survival	Toxicity
**Before HAART Era**									
Nobler et al. [10]	1987	USA	31	106	Skin lesion:20 Gy (3 Gy/F) Visceral lesion:16–20 Gy (1.6–2 Gy/F)	36/106	18/106	8.7 months	NA
Chak et al. [11]	1988	USA	24	80	20 Gy (5 × 2 Gy/w)	87%	NA	7 months	4/9 (oral lesion) G1+ epithelitis and mucositis
Cooper et al. [12]	1990	USA	129	226	30 Gy (5 × 2 Gy/w) (*n* = 186) 1 × 8 Gy (*n* = 40)	NA	68%	9.6–11.8 months	NA
De Wit et al. [9]	1990	The Netherlands	31	74	1 × 8 Gy	34%	8%	65% alive (FU unknown)	G1-2 Epithelitis = 13% G1-2 mucositis = 13%
Berson et al. [13]	1990	USA	187	375	1.8–2.5 Gy × 10–15 F 1 × 6–8 Gy	96% 93% (1 × 8 Gy)	65% 68% (1 × 8 Gy)	6 months	G2 = 32% G3 = 2%
Ghabrial et al. [14]	1992	USA	26 16	31 18	1 × 8 Gy 15–36 Gy	100% 100%	32% 24%	9.2 months	G1+ = 19%
Stelzer et al. [8]	1993	USA	14	24 24 23	1 × 8 Gy 20 Gy (5 × 2 Gy/w) 40 Gy (5 × 2 Gy/w)	88% 88% 100%	50% 79% 83%	86% at ≈ 10 months	G1+ = 12.5% G1+ = 46% G1+ = 55%
Kirova et al. [15]	1998	France	643 Skin (*n* = 576) Oral (*n* = 62) Eyelid, conjunctiva, lip, and Genital (*n* = NA)	6777 Skin (*n* = 6111) Oral (*n* = 115) Eyelid, conjunctiva, lip, and Genital (*n* = 551)	Skin lésion:20 Gy + 10 Gy (4 × 2.5 Gy/w) Oral lesion:15.2 Gy +/− 15.2 Gy (4 × 1.9 Gy/w) Eyelid, conjunctiva, lip, and Genital lesion: 10 Gy + 10 Gy (4 × 2.5 Gy/w)	92% 100% 89%	66% 17.8% 26%	8.2 months	Epithelitis G1 = 7.3% G2 = 69.3% G3 = 23.4%
Harrison et al. [6]	1998	UK	57	596	4 × 4 Gy 1 × 8 Gy	96% 91%	78% 81%	17 months	NA
Wulf et al. [16]	2021	Denmark	9 53 32	74 2066 165	3 × 2 Gy(skin) 3 × 4 Gy(skin) 3 × 4 Gy(mucous)	100% 100% 100%	70% 93% 91%	NA	NA
**HAART Era**									
Singh et al. [7]	2008	South Africa	60	35 30	24 Gy (2 Gy/F) 20 Gy (5 × 4 Gy)	91% 96%	57% 56%	5.5 months (median survival)	57% 33% (Acute epithelitis)
Donato et al. [17]	2013	Italia	15	38	20–36 Gy (2 Gy/F)	100%	83.8%	51.4 months (mean)	G1 epithelitis = 21%
Tsao et al. [18]	2016	Canada	3	8	1 × 8 Gy; 5 × 4 Gy or 10 × 3 Gy	100%	25%	NA	G1 epithelitis

^1^ CR: Complete response; F: fraction; HAART: Highly Active AntiRetroviral Therapy; NA: Not available; RT = Radiotherapy.

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
