# Peer review of "The Role of Radiotherapy in Treating Kaposi’s Sarcoma in HIV Infected Patients"

_cancers, 2022, doi:10.3390/cancers14081915_

Round 1
Reviewer 1 Report
The authors provided a comprehensive summary of what we know about the effectivity of radiotherapy in KS treatment. The clinical studies are well described but I missed that there is no mention about what KS is.
I recommend that they write a paragraph about what kind of disease KS is and what causes it. KS is known to be caused by Human Hepresvirus 8 (Kaposi's sarcoma-associated herpesvirus) infection and it is derived from infected lymphatic endothelial cells...etc. Also, a brief description of how KS lesions look like would be helpful for the readers.
Minor:
Move the head of the Table “HAART Era” to the next page
Line 114/115 …for skin lesions, time to recurrence…
Minor spell check is required.
Author Response
Reviewer#1: The authors provided a comprehensive summary of what we know about the effectivity of radiotherapy in KS treatment. The clinical studies are well described but I missed that there is no mention about what KS is.
Reviewer#1: I recommend that they write a paragraph about what kind of disease KS is and what causes it. KS is known to be caused by Human Hepresvirus 8 (Kaposi's sarcoma-associated herpesvirus) infection and it is derived from infected lymphatic endothelial cells...etc. Also, a brief description of how KS lesions look like would be helpful for the readers.
Authors: In the introduction, we have added the following paragraph on the pathophysiology and clinical aspect of Kaposi's disease: Kaposi sarcoma (KS) is an endothelial cells malignancy caused by human herpesvirus 8 (HHV-8), also known as Kaposi's sarcoma (KS) associated herpesvirus. KS can affect the skin, lymph nodes and/or viscera. The skin lesions may present in different clinical forms such as ulcerations, patches, plaques or colored nodules (red, bluish or sometimes brown) single or multiple and are frequently associated with edema. These lesions can be painful and cause bleeding or infection such as cellulitis. Histologically, KS is characterized by a combination of an inflammatory infiltrate, endothelial cells and spindle-shaped cells. In HIV-infected patients, KS is classified by the Centers for Disease Control and Prevention (CDC) as AIDS-defining malignancy. With the development of HAART, the incidence of KS has greatly decreased in the US, from a peak of about 47 per million in the 1990s to 6 per million currently.
Minor:
Reviewer#1: Move the head of the Table “HAART Era” to the next page
Authors: We move the head of the Table “HAART Era” to the next page.
Reviewer#1: Line 114/115 …for skin lesions, time to recurrence…
Authors: we have corrected the mistakes
Reviewer#1: Minor spell check is required.
Authors: We asked to Dr. Marc Bollet, who is an English native speaker, to proofread and correct the English revised version of the manuscript. We have added him to the acknowledgements paragraph.
Reviewer 2 Report
Laurent Quéro, Romain Palich and Marc-Antoine Valantin present a review article discussing the role of radiotherapy in treating KSHV/HIV-infected patients as a holistic strategy to prevent KS severe pathogenesis. The inclusion of research made in different countries from 1987 to 2016 summarizes the overall main point of the authors to consider radiotherapy as a therapeutic against KS, they selected 10 publications and used them to support their conclusions in favor of radiotherapy for KS treatment. The manuscript, however has serious flows, major: 1. They need to introduce Kaposi’s sarcoma, the etiology, the demographics, the HIV-related and work on separating them from the AIDS-related (which seems to be the main focus of the manuscript, in a weird combination with HIV-related). 2. They seem to ignore the latent aspect of the herpesvirus cycle. Specifically a gamma-herpesvirus as KSHV. Standard therapies for these kinds of virus-associated-diseases involve approaches such as treating the underlying immunodeficiency, cytotoxic chemotherapy, and immunologic antitumor therapy that are not mentioned. Novel therapy approaches include specific immune therapy and anti-angiogenesis approaches, all these are missing in the article and are essential to consider if a holistic approach is being proposed. 3. They claim that radiotherapy has a high success 80-90% response rates. However, they are ignoring that treating spontaneous reactivation of herpes will produce what they call: “.. possible risk of the development of late skin sequelae and the possibility of recurrence.” The authors do not mention the long-term success rate of the therapy or the evidence that the latent episome is not existent or will not reactivate. minor: 1 . Authors need to be aware that Extensive editing of English language and style is required. 2. The authors should reconsider the title: “Place of radiotherapy in the treatment of Kaposi’s sarcoma in HIV infected patients”, maybe rewording as: The role of radiotherapy in treating Kaposi’s sarcoma in HIV infected patients. Therefore I can not recommend Laurent Quéro, et.al request for publication in Cancers.Author Response
Reviewer#2: Laurent Quéro, Romain Palich and Marc-Antoine Valantin present a review article discussing the role of radiotherapy in treating KSHV/HIV-infected patients as a holistic strategy to prevent KS severe pathogenesis. The inclusion of research made in different countries from 1987 to 2016 summarizes the overall main point of the authors to consider radiotherapy as a therapeutic against KS, they selected 10 publications and used them to support their conclusions in favor of radiotherapy for KS treatment.
Reviewer#2 The manuscript, however has serious flows,
major: 1. They need to introduce Kaposi’s sarcoma, the etiology, the demographics, the HIV-related and work on separating them from the AIDS-related (which seems to be the main focus of the manuscript, in a weird combination with HIV-related).
Authors: In the introduction, we have added a paragraph on the pathophysiology and clinical aspect of Kaposi's sarcoma
Reviewer#2. They seem to ignore the latent aspect of the herpesvirus cycle. Specifically, a gamma-herpesvirus as KSHV. Standard therapies for these kinds of virus-associated-diseases involve approaches such as treating the underlying immunodeficiency, cytotoxic chemotherapy, and immunologic antitumor therapy that are not mentioned. Novel therapy approaches include specific immune therapy and anti-angiogenesis approaches, all these are missing in the article and are essential to consider if a holistic approach is being proposed.
Authors: This article focused on the treatment of Kaposi's disease by radiotherapy, is part of a special issue of the journal Cancers which is composed of 6 articles and which is dedicated to Kaposi's disease. This explains why our manuscript does not address the themes mentioned by reviewer #2 which are covered in the other articles of this special issue.
We are agree with the comments of the reviewer #2, radiotherapy is indeed a loco-regional treatment which is part of a multimodal management strategy of Kaposi's disease, its effectiveness concerns only the control of the disease in the irradiated territories. We have made this obvious clarification in the introduction of the manuscript
Reviewer#2. They claim that radiotherapy has a high success 80-90% response rates. However, they are ignoring that treating spontaneous reactivation of herpes will produce what they call: “.. possible risk of the development of late skin sequelae and the possibility of recurrence.” The authors do not mention the long-term success rate of the therapy or the evidence that the latent episome is not existent or will not reactivate.
Authors: Due to the short follow-up time of patients in most studies, the long-term recurrence rate in and out of the radiation field is not known
minor:
Reviewer#2 . Authors need to be aware that Extensive editing of English language and style is required.
Authors: We asked to Dr. Marc Bollet, who is an English native speaker, to proofread and correct the English revised version of the manuscript. We have added him to the acknowledgements paragraph.
Reviewer#2. The authors should reconsider the title: “Place of radiotherapy in the treatment of Kaposi’s sarcoma in HIV infected patients”, maybe rewording as: The role of radiotherapy in treating Kaposi’s sarcoma in HIV infected patients. Therefore, I can not recommend Laurent Quéro, et.al request for publication in Cancers.
Authors: As suggested by reviewer #2, we change the title of the manuscript: “Place of radiotherapy in the treatment of Kaposi’s sarcoma in HIV infected patients” with “The role of radiotherapy in treating Kaposi’s sarcoma in HIV infected patients”
Round 2
Reviewer 2 Report
The authors provided substantial changes to the manuscript in this revised version that improved the overall document. Some of my questions about the latent state of the disease were included in the previous review report and were not addressed anywhere. Still, the manuscript has some improvements that would be of interest to the readers as a part of a special issue.